# On the Mechanism of Membrane Permeabilization by Tamoxifen and 4-Hydroxytamoxifen

**DOI:** 10.3390/membranes13030292

**Published:** 2023-02-28

**Authors:** Julia Ortiz, José A. Teruel, Francisco J. Aranda, Antonio Ortiz

**Affiliations:** Departamento de Bioquímica y Biología Molecular-A, Facultad de Veterinaria, Universidad de Murcia, E-30100 Murcia, Spain

**Keywords:** tamoxifen, 4-hydroxytamoxifen, phospholipid membranes, membrane permeabilization

## Abstract

Tamoxifen (TMX), commonly used in complementary therapy for breast cancer, also displays known effects on the structure and function of biological membranes. This work presents an experimental and simulation study on the permeabilization of model phospholipid membranes by TMX and its derivative 4-hydroxytamoxifen (HTMX). TMX induces rapid and extensive vesicle contents leakage in phosphatidylcholine (PC) liposomes, with the effect of HTMX being much weaker. Fitting of the leakage curves for TMX, yields two rate constants, corresponding to a fast and a slow process, whereas in the case of HTMX, only the slow process takes place. Interestingly, incorporation of phosphatidylglycerol (PG) or phosphatidylethanolamine (PE) protects PC membranes from TMXinduced permeabilization. Fourier-transform infrared spectroscopy (FTIR) shows that, in the presence of TMX there is a shift in the ν_CH2_ band frequency, corresponding to an increase in *gauche* conformers, and a shift in the ν_C=O_ band frequency, indicating a dehydration of the polar region. A preferential association of TMX with PC, in mixed PC/PE systems, is observed by differential scanning calorimetry. Molecular dynamics (MD) simulations support the experimental results, and provide feasible explanations to the protecting effect of PG and PE. These findings add new information to explain the various mechanisms of the anticancer actions of TMX, not related to the estrogen receptor, and potential side effects of this drug.

## 1. Introduction

Tamoxifen (TMX) is currently used for the complementary endocrine treatment of patients with breast cancer [1,2]. Being considered a prodrug, TMX is transformed into 4-hydroxytamoxifen (HTMX) and endoxifen in the organism, all these compounds behaving as antiestrogens [2], blocking estrogen receptors [3,4], and thus exerting their main pharmacological actions. Nevertheless, in addition to receptor binding, the preferential partitioning of TMX and HTMX into cellular membranes, due to their highly hydrophobic character, results in additional alterations of membrane structure and function [5]. The interactions of TMX and HTMX with phosphatidylcholine (PC) model membranes have been studied in a number of works [6,7,8,9,10,11,12], providing details on their effects on membrane fluidity, order, hydration, and other physicochemical parameters. In addition, one recent work has reported on the interactions of TMX and HTMX with a phosphatidylethanolamine (PE) system, the second most important phospholipid in biological membranes [13], showing that these two drugs facilitate the formation of the inverted hexagonalH_II_ phase [14].

The mechanism of drug-induced phospholipid membrane permeabilization, strongly depends on the physicochemical properties of the target membrane [15,16]. Many works have described membrane permeabilization in model vesicles or liposome systems of various compositions, induced by different compounds, among which cytotoxic amphipathic helices [17], apoptotic peptides [18], and bacterial glycolipids [19,20], or lipopeptides [21] can be cited. Within this context, the hemolytic activity of TMX has been described [22], and it has also been briefly reported that TMX can permeabilize model phospholipid vesicles [9,10,11]. Thus, it seems evident that the interaction of TMX with biological membranes not only alters their physicochemical properties, but also compromises their barrier function, an issue which deserves more profound study. The central goal of the current study is to show that TMX and HTMX can perturb the structure and function of phospholipid membranes, leading to membrane permeabilization, which may be behind some of their pharmacological actions or side effects, not related to receptor binding. This is supported by previous data, showing that other drug-like molecules also alter model phospholipid membranes, in addition to receptor binding. Thus, fenamates, used as COX inhibitors, have been shown to alter POPC membrane’s structure, dynamics, and fluidity [23], these authors suggesting that these actions may cause potential side effects. Other NSAIDs, such as diclofenac and naproxen, have also been shown to affect the permeability and structure of model lipid bilayers, which various authors have suggested may explain their cardiotoxicity [24,25]. In this work, in order to obtain further details on the mechanism of membrane permeabilization induced by TMX and HTMX, we carried out detailed kinetic experiments to monitor contents leakage in model phospholipid unilamellar vesicles, of various lipid compositions. A dissimilar action of TMX and HTMX on vesicle membrane permeabilization was found, as well as an influence of membrane composition on the rate and extent of leakage. These results were complemented with Fourier-transform infrared spectroscopy (FTIR) and differential scanning calorimetry (DSC) data, as well as molecular dynamics (MD) simulations, providing a model for the molecular mechanism of TMXinduced phospholipid membrane permeabilization.

## 2. Materials and Methods

### 2.1. Materials

1-Palmitoyl-2-oleoyl-*sn*-glycero-3-phosphocholine (POPC), 1-palmitoyl-2-oleoyl-*sn*- glycero-3-phosphoethanolamine (POPE), 1-palmitoyl-2-hydroxy-*sn*-glycero-3- phosphoglycerol (POPG), 1,2-dilauroyl-*sn*-glycero-3-phosphoethanolamine (DLPE), 1,2-dipalmitoyl-*sn*-glycero-3-phosphoethanolamine (DPPE), 1,2-dimyristoyl-*sn*- glycero-3-phosphocholine (DMPC), and 1,2-distearoyl-*sn*-glycero-3-phosphocholine (DSPC) were purchased from Avanti Polar Lipids Inc. (Birmingham, AL). TMX, HTMX, and 5(6)-carboxyfluorescein (CF) (99% by HPLC) were from Sigma-Aldrich (Madrid, Spain). All the other reagents were of the highest purity available. Inorganic salts and buffers were of analytical grade. Purified water was deionized in a Milli-Q equipment from Millipore (Millipore, Bedford, MA), and had a resistivity of ca. 18 MΩ. Stock solutions of the various phospholipids and drugs were prepared in chloroform and stored at −80 °C. Phospholipid phosphorus was determined according to the method of Böttcher [26]. The buffer used through the work was 150 mM NaCl, 5 mM Hepes, pH 7.4, unless otherwise stated. Water, and all the buffer solutions used in this work, was filtered through 0.2 μm filters prior to use. The osmolarities of all the buffers and solutions were checked using a Osmomat 030 osmometer (Gonotec, Berlin, Germany).

### 2.2. Vesicle Preparation for Membrane Permeabilization

The phospholipid vesicles to be used in the CF leakage experiments were prepared by mixing the appropriate amounts of phospholipids in organic solvent. The solvent was gently evaporated under a stream of dry N_2_ to obtain a thin film at the bottom of a small glass tube. Any final traces of solvent were removed by a further 3 h desiccation under high vacuum. To the dry samples, 1 mL of a buffer, containing 75 mM CF, 5 mM Hepes, pH 7.4, was added (final phospholipid concentration 10 mM), and multilamellar vesicles were formed by vortexing the mixture. Large unilamellar vesicles (LUV) were prepared by repeated extrusion (11×) of the multilamellar vesicles through two stacked 0.1 μm polycarbonate filters, using a LiposoFast device (Avestin, Ottawa, ON). Vesicles were separated from non-encapsulated material by gel filtration, on Sephadex G-50, using 150 mM NaCl, 5 mM Hepes, pH 7.4, as elution buffer. Under these conditions LUV eluted with the void volume, whereas free CF was retained. LUV preparations were collected, and lipid concentration was determined by phosphate analysis, as indicated above. Osmolarity was adjusted with NaCl, to match with that of the elution buffer described above.

### 2.3. Fluorescence Measurements

Fluorescence measurements were carried out in a PTI Quantamaster spectrofluorometer (Photon Technology, Birmingham, NJ). Quartz cuvettes, with path lengths of 1 × 1 cm, were used. Release of CF from LUV was measured in the same buffer indicated above, at excitation and emission wavelengths of 430 and 520 nm, respectively. TMX or HTMX were added to the buffer from stock solutions in DMSO. The volume added was always less than 5% of the total buffer volume and, in any case, fluorescence intensities were corrected for these small dilutions. It was carefully checked that the same volumes of DMSO alone did not induce any leakage. All measurements were performed in a thermostated stirred cell holder, at 25 °C. Maximum leakage (100%) was established by dissolving the liposomes with 0.5% Triton X100, which provoked the complete release of CF to the external medium, and the percentage of CF leakage was calculated as:$$\%\;CF\;Leakge=\frac{\left(F_{t}-F_{i}\right)\cdot 100}{F_{d}-F_{i}}$$
where *F_t_* is the fluorescence at a given time after drug injection, *F_i_* is the initial fluorescence, and *F_d_* is the maximum fluorescence obtained after addition of the detergent Triton X100. The fluorescence of CF has been shown to depend linearly on concentration up to 0.002 M [27], a concentration which is far above the CF concentration reached at *F_d_*, i.e., disrupting the vesicles with detergent makes the % of leakage directly proportional to CF fluorescence [28]. The CF leakage curves shown in Figures 2 and 3 could be well fitted to two exponentials according to the equation:(1)$$\%\;CF\;Leakage=100-a\cdot e^{-k_{1}\cdot t}-b\cdot e^{-k_{2}\cdot t}$$
where *k*_1_ and *k*_2_ are the rate constants, and a and b are their relative proportions (%). These two rate constants correspond to two different physical processes of CF permeation across the membrane, as will be shown below.

### 2.4. Fourier-Transform Infrared Spectroscopy

Multilamellar vesicles for FTIR were prepared, essentially as described before [29], in 50 μL of the same buffer, prepared with D_2_O. CaF_2_ windows (25 × 2 mm), with 25 μm Teflon spacers, were used. Infrared spectra were acquired in a Nicolet 6700 FTIR spectrometer (Madison, WI, USA), collecting 64 interferograms, with a nominal resolution of 2 cm^−1^. The equipment was continuously purged with dry air, in order to minimize the contribution peaks of atmospheric water vapor, and the holder was thermostated using a Peltier device (Nicolet Proteus System). Spectra were collected at 2 °C intervals, allowing 5 min equilibration between temperatures. The D_2_O buffer spectra, taken at the same temperatures, were subtracted interactively using either the Omnic (version 7.3, Thermo Scientific, Waltham, MA, USA), or Grams (version 7.02, Thero Scientific, Waltham, MA, USA) software.

### 2.5. High Sensitivity Differential Scanning Calorimetry

Multilamellar vesicles for DSC measurements were prepared by the dry film hydration method, essentially as described previously [11]. Briefly, the appropriate amounts of phospholipids and TMX were mixed in chloroform, and the organic solvent was gently evaporated using dry N_2_, to obtain a thin film at the bottom of a glass tube. Any remaining traces of solvent were removed by a further 3 h desiccation under high vacuum. To the dry samples, 0.5 mL of a buffer containing 150 mM NaCl, 0.1 mM EDTA, 10 mM Hepes, pH 7.4, was added, and vesicles were formed by vortexing the mixture at 50 °C (for the DLPE/DSPC mixture) or 70 °C (for the DMPC/DPPE mixture) (final phospholipid concentration 2 mM). Experiments were performed in a PEAQ-DSC equipment (Malvern Pananalytical, Malvern, UK), at 4 °C min^−1^ heating scan rate. Three consecutive heating scans were carried out for each sample, the last one being taken for analysis.

### 2.6. Molecular Dynamics

The molecular structure of TMX is available in the PubChem Substance and Compound database [30], through the unique chemical structure identifier 2733526. All MD simulations were performed using GROMACS 5.0.7 and 2018.1 [31], at the Computational Service of the University of Murcia, Spain. CHARMM36 force field parameters for POPC, POPE, POPG, TMX, water, Cl^−^, and Na^+^ were obtained from CHARMM-GUI [32,33,34]. The simulated systems were POPC/TMX (15.9, 25.6, and 33.3 TMX mol%), POPC/POPE/TMX (24:8:11; i.e., 25.6 mol% TMX), and POPC/POPG/TMX (24:8:11; i.e., 25.6 mol% TMX). The membrane bilayers were formed by two leaflets, oriented normal to the *z*-axis, with a total of 128 molecules of lipids, with 24, 44, or 64 molecules of TMX, and a water layer containing a total of 12,800 water molecules, 45 sodium ions, and chloride ions to neutralize the charge of the simulated system. For water molecules, the TIP3P model was used, which has been proven to be useful for membrane models with the CHARMM36 force field in our previous study [35,36,37]. The initial membrane structures were built with the aid of the Packmol software [38]. 

All systems were simulated using the NpT-ensemble at 298.15 K. Pressure was controlled semiisotropically, at a pressure of 1 bar and compressibility of 4.5 × 10^−5^ bar^−1^. The cutoffs for van der Waals and short-range electrostatic interactions were 1.2 nm, and a force switch function was applied between 1.0 and 1.2 nm [37]. The time step was 2 fs. Simulations were initiated by a 20 ns run, which was sufficient to achieve a rapid initial equilibration (Appendix A), using the V−rescale thermostat and the Berendsen barostat [39], followed by a 180 ns run using the Nose–Hoover thermostat [40] and the Parrinello–Rahman barostat [41]. Graphical representations were produced with PyMOL 2.3.0 [42]. All systems were simulated in triplicate, yielding similar results, and a representative one was chosen for analysis. Analysis of the trajectories was performed over the last 60 ns, using the GROMACS tools and homemade scripts.

## 3. Results

The hydrophobic nature of TMX and its derivatives makes these compounds prompt to partitioning into biological membranes, which results in the perturbation of their structural and functional properties. Here, we carried out an experimental and molecular dynamics (MD) study on the permeabilization of model phospholipid membranes by TMX and HTMX (Figure 1), and on its modulation by membrane lipid composition. Given that phosphatidylcholine is the most abundant phospholipid species in biological membranes [43], a synthetic POPC species was used as the main component of the vesicles, with the advantage that its gel to liquid–crystalline phase transition takes place below 0 °C, thus being fluid at 25 °C [44]. The TMX concentrations used in this work correspond to drug/phospholipid ratios from 0.125:1 up to 2.5:1, a range previously used by other authors in similar studies with TMX [10], or even much below those used in similar studies on the interaction with model membranes of other compounds such as, for instance, NSAIDs [24,25] or monoterpenes [45]. These drug concentrations are frequently used in studies with model membranes [10,24,25], and result in drug membrane levels higher than those reached with the pharmacological doses, but are required in order to observe measurable effects with current biophysical techniques. In the case of TMX, the habitual pharmacological doses oscillate around 20 mg/day, reaching plasma concentrations of up to 230 ng mL^−1^ (0.62 μM) [46]. In this respect, it has to be taken into consideration that: (i) these are long-term treatments which result in the progressive accumulation of TMX in body cell membranes [46]; and (ii) the differential partitioning of TMX, depending on membrane composition [12], should result in a non-homogeneous distribution of the drug within the body cell membranes. Thus, the use of higher drug concentrations in model membrane studies does not limit their biological relevance since, due to the several factors mentioned above, it is feasible that local drug membrane concentrations approaching the lowest ratios used in this work can also be reached in vivo. 

### 3.1. Drug-Induced Membrane Permeabilization

Figure 2 shows the curves of CF leakage from POPC LUV, resulting upon addition of TMX or HTMX from outside. TMX induced rapid and extensive vesicle contents leakage in a concentration-dependent manner (Figure 2a). It can be seen that TMX concentrations as low as 2.5 μM (1:10 drug/lipid ratio) already gave rise to a measurable membrane permeabilization, which rapidly increased upon raising drug concentration, being quite fast at 50 μM. It has to be mentioned that, all the curves reached 100% CF leakage if allowed to proceed for longer times. Interestingly, the effect of HTMX within the same concentration range (Figure 2b) was almost negligible, as compared to TMX. Thus, at the maximum concentration assayed (50 μM), TMXinduced CF leakage approached 100% after 180 s, whereas HTMX-induced leakage barely reached 10% of the maximum. Nevertheless, it should be remarked that, even the lowest concentration of HTMX assayed (2.5 μM) also gave rise to a much slower, but still measurable, permeabilization of the membrane.

The leakage curves shown in Figure 2 could not be fitted to one exponential, but to the sum of two exponentials, according to Equation (1). This fitting procedure yielded the values of two rate constants (k_1_ and k_2_), as well as their relative proportions (Table 1). It can be observed that two processes with very different rates were taking place at the same time. For either drug, the values of the k_1_ constant ranged between 0.64 × 10^−4^ and 129 × 10^−4^ s^−1^, corresponding to the slow process, whereas k_2_ ranged from 0.020–0.056 s^−1^, indicating a faster CF leakage. For TMX, the proportion of the k_2_ component, which was very low at low drug concentrations (2.0% at 2.5 μM or 0.125:1 drug/lipid ratio), increased upon increasing TMX concentration, reaching 58% at 50 μM TMX (2.5:1 drug/lipid ratio). 

These data also indicated that the slow CF leakage shown in Figure 2b, induced by HTMX, corresponded to the slow process (k_1_) (ranging 98.4–99.6%), with essentially no contribution of the faster (k_2_) component (just 1.6% at 50 μM). A plot of the fractional contribution of the k_2_ rate constant as a function of drug concentration is shown in Figure 3. It was observed that the proportion of the k_2_ component increased, upon increasing TMX concentration, in a nonlinear manner, with an inflection point starting around 10 μM TMX (0.5:1 drug/lipid ratio). This sigmoidal dependence with TMX/phospholipid ratio was similar to that published before for the plot of the % of calcein leakage induced by TMX [10], which also reached a plateau around a 1:1 drug/lipid ratio. Notably, for HTMX, the proportion of the slow component, k_2_, remained essentially constant within this range of drug concentrations. 

An important aim set in this work, was to evaluate the influence of membrane phospholipid composition on TMX and HTMX-induced vesicle contents leakage. For that purpose, three different membrane compositions were selected, namely, pure POPC, POPC/POPG (5:1.7), and POPC/POPE (5:1.7). The incorporation of POPG, an anionic phospholipid [47], allowed the examination of the effect of the drugs on a negatively charged membrane, as well as the influence of a phospholipid voluminous headgroup. On the other hand, POPE was selected due to the biological relevance of phosphatidylethanolamine, the second most abundant phospholipid in biological membranes, in addition to its rich lipid polymorphism [43], having in this case a small headgroup. Drug-induced CF leakage curves obtained for these systems are shown in Figure 4.

A full range of drug concentrations was assayed (as shown in Figure 2), but only the results for 17.5 μM are shown, for the sake of simplicity. As shown in Figure 4a, POPG exerted a protective effect, decreasing the % of leakage more than 10% after 180 s. Interestingly, inclusion of POPE resulted in a negligible TMX-induced permeabilization of the membrane, as compared to the other compositions, an effect which was observed at this concentration and all other concentrations assayed, up to 50 μM. The inset shows a close-up view of the trace obtained for POPC/POPE vesicles, showing that, although slow, membrane permeabilization occurred. Again, the effect of HTMX (Figure 4b) was much weaker for all the compositions studied, but the same pattern as for TMX was observed, from a qualitative point of view.

The curves shown in Figure 4 were also fitted to Equation (1), to obtain the values and proportions of k_1_ and k_2_, as above, and the results are shown in Table 2. For TMX, incorporation of negatively charged POPG into POPC membranes resulted in a minor reduction in the fast (k_1_) component and its contribution, as compared to the results obtained for pure POPC vesicles. However, in the presence of POPE, the curves fitted well to just one exponential, resulting in the disappearance of the faster (k_2_) component of leakage, and the value of k_1_ was also reduced. The values and contributions of k_1_ and k_2_, for HTMXinduced CF leakage, were essentially not affected by incorporation of POPG, and incorporation of POPE did not change the value of k_1_.

In light of the weak effects of HTMX on membrane permeabilization described above, the rest of our study was focused on understanding the molecular mechanism of TMX’s action. From similar leakage curves to those shown in Figure 4a, obtained at various TMX concentrations, the initial rates of leakage were determined from the tangents to the curves at time zero (Figure 5). For pure POPC, the onset concentration for membrane permeabilization, determined from the inflection point of the curve, was ca. 10 μM, corresponding to a 1:2 TMX/POPC ratio, a value in good agreement with the results shown in Figure 3. It can be seen that introduction of POPG did not essentially modify this parameter but, interestingly, in the case of POPC/POPE membranes, the dependence of the initial rate of leakage with TMX concentration was linear (no inflection) within this range of concentrations.

### 3.2. Fourier-Transform Infrared Spectroscopy

FTIR was used to obtain information on the effect of TMX on the hydrophobic (CH_2_ groups) and hydrophilic (C=O groups) regions of the membrane, for the various systems under study (Table 3). The symmetric stretching band of the phospholipid methylene groups (ν_CH2_) is particularly useful, since it does not overlap with bands from other groups, and monitors changes in the conformational disorder of the acyl chains [48]. A general trend observed was that, in the presence of 25 mol% TMX, the maximum frequency of the ν_CH2_ symmetric stretching band of the phospholipids (that appears around 2850 cm^−1^) was shifted to higher values, with a weaker effect in the case of POPC/POPG membranes (1 cm^−1^ shift), indicating an additional disordering of the fluid bilayer. 

The effect on the polar region of the membrane was evaluated by analyzing the ν_C=O_ stretching band of the phospholipids, around 1730 cm^−1^. It was observed that, for all cases, inclusion of TMX into the membrane shifted the ν_C=O_ band maximum toward higher values, indicating a dehydration of the polar region of the bilayer, caused by the drug [49,50]. Again, a smaller shift was observed in the case of the membranes containing POPG.

### 3.3. Differential Scanning Calorimetry

In an effort to shed light on the particularly strong effect of PE, DSC experiments of TMX incorporated into various mixed PC/PE systems were carried out (Figure 6). Two systems showing solid-phase separation were chosen [51], namely DLPE/DSPC (1:1) and DMPC/DPPE (1:1), such that in one case PE was the most fluid component, whereas in the other it was PC. It could be observed that these mixtures showed two well separated phase transitions, corresponding to phases rich in each one of the components, but, as expected, were not as cooperative as those observed for the pure components alone, since it is not possible to achieve a complete separation within the membrane, as described previously [51]. Nevertheless, the separation of the two transitions was good enough to allow monitoring of the effect of TMX on each particular component.

The presence of increasing concentrations of TMX in DLPE/DSPC membranes, (Figure 6a) progressively shifted the high-melting component (DSPC) toward lower values, with the appearance of a high temperature peak at 20 mol% TMX, and widened the transition. However, the T_m_ of the low-melting component (DLPE) was just slightly increased, and the transitions were sharper in the presence of TMX. For the DMPC/DPPE system (Figure 6b), incorporation of TMX clearly affected the low-melting component (DMPC), which was progressively widened, and almost disappeared at 20 mol%, whereas the high-melting component (DPPE) slightly shifted toward lower temperatures, and not only did it not widen but it even became sharper in the presence of TMX. 

### 3.4. Molecular Dynamics Simulations

Molecular dynamics (MD) simulations in membranes [52] were carried out, to obtain information on the interaction of TMX with POPC bilayers, in the absence and presence of POPE or POPG. To obtain a proof of convergence of the simulation, the area per lipid can be used as a trustable parameter of the lipid bilayer. The average area per lipid was calculated from the cross-sectional area of the simulation box (xy-plane) divided by the number of lipid molecules in one leaflet of the bilayer membrane. It was observed (Appendix A) that this parameter converged, and remained essentially constant within the time range used for the various membrane compositions, indicating that all these systems reached equilibrium. First of all, the electrostatic and hydrophobic energies of interaction between POPC and TMX in POPC membranes containing increasing concentrations of TMX were calculated (Figure 7). Electrostatic interactions between groups were calculated as short-range Coulomb energy interactions, and hydrophobic interactions as short-range Lennard–Jones energy interactions. It was observed that the electrostatic interactions were weak and did not change upon increasing the drug concentration. However, hydrophobic interactions progressively increased, up to a maximum at 15.9 mol% TMX, due to the insertion of the drug into the membrane, and then, interestingly, progressively decreased upon further increasing TMX concentration. This finding most likely indicated a predominant TMX–TMX interaction over TMX–POPC, compatible with drug clustering.

Analysis of mass density profiles (Figure 8) showed that TMX could be found all along the whole lipid phase, but was mainly located at the center of the bilayer, in agreement with previous data on fluid DPPC membranes [11]. No relevant differences were found between the various compositions under study.

The phospholipid–phospholipid energies of interaction and hydrogen bonding calculated for the various systems under study (Figure 9), provided data supporting the protecting effect of POPG and POPE on TMXinduced membrane permeabilization. Energies were calculated as described above, and a hydrogen bonding was defined based on a cutoff of 35° for the hydrogen donor–acceptor angle, and a donor–acceptor distance ≤ 0.35 nm. The calculations showed that, in the case of POPC membranes, electrostatic interactions were repulsive and hydrophobic ones were attractive. Remarkably, insertion of POPG or POPE within the membrane changed this scenario, by introducing attractive electrostatic interactions, without significantly modifying hydrophobic attraction. Furthermore, the presence of PG or PE also allowed phospholipid–phospholipid hydrogen bond formation, which was not possible for POPC alone membranes.

MD did not show significant differences in hydrogen bonding between the phospholipids’ C=O groups and water molecules, in the absence and presence of TMX, for either composition (not shown). Furthermore, hydrogen bond formation between the phospholipids’ C=O groups and water molecules, for TMX incorporated in membranes of various compositions, was also examined, and we did not observe any significant differences in water hydrogen bonding caused by membrane composition (Appendix A). The same trend was obtained for the proportion of *gauche* conformers (Appendix A), which was calculated from the probability distribution of the dihedral angles of the lipid acyl chains.

## 4. Discussion

TMX and its derivatives, frequently used for the complementary management of breast cancer, bind estrogen receptors, which constitute their main cellular targets [3,4]. Nevertheless, it has also been proposed that other mechanisms, not related to the receptor, may be involved in their pharmacological activity [53]. The strong lipophilic character of TMX and HTMX, facilitates their partition into phospholipid membranes, which alters membrane structure and function [5]. In fact, the partitioning of these two drugs into various membranes was determined early, and it was found to be of the same order of magnitude, but being affected by membrane composition and temperature [12], indicating that both compounds have a large tendency to partition into hydrophobic phospholipid bilayers. The TMX-related membrane actions could explain some of the pharmacological activities, or concomitant side effects, of these drugs, not related to estrogen receptor binding. This is the case for other drug-like compounds such as fenamates [23], or NSAIDS [24,25], that have also been shown to alter membrane structure and function, in addition to receptor binding, which may explain some of their toxic actions. Accordingly, a number of experimental works have addressed the study of the interaction of TMX and HTMX with biological membranes. TMX has been shown to widen and shift the gel to liquid–crystalline phase transition of PC model membranes [6,7,8,9,10,11], to alter membrane fluidity [7,54], to induce stomatocyte formation in human erythrocytes [55], and was briefly reported to permeabilize model phospholipid vesicles [9,10,11].

With the above scenario in mind, an experimental and MD work was performed, to deepen our understanding of the mechanism of TMX-induced membrane permeabilization, extending the study to HTMX, one of TMX’s principal metabolites, and with an emphasis on kinetic characteristics and membrane composition. Negatively charged PG was used because, although being a minor constituent in most biological membranes, in light of its physicochemical and biological properties, this phospholipid displays a special biological relevance [56]. PG is mainly found in bacterial cell membranes, and in eukaryotic cell membranes it is relatively more abundant in the mitochondrial membrane, which has been shown to be a target for some bioactive compounds [57,58]. In addition, PG can act as a signal molecule, inhibiting various inflammatory responses [59,60,61], or even suppressing viral infection [62]. Furthermore, the use of PG in this work has allowed investigation of the role of a voluminous phospholipid headgroup, on TMX-induced leakage. PE was selected as a component of the model membrane systems because of its biological relevance as the second major phospholipid of most living cell membranes [13,43], playing a special role in cancer cell membranes [63], and presenting a rich lipid polymorphism [64].

We found that, for pure POPC membranes, the effect of TMX is the summation of a fast leakage component, taking place in the range of seconds, and a slow one, which completes within minutes–hours, depending on drug concentration. Previous authors have also described biphasic TMX-induced contents leakage in vesicles of a different composition [9], reporting k_1_ and k_2_ rate constants very similar to those found in our experiments. However, we extended the study to higher TMX concentrations, and showed that raising the TMX/POPC ratio progressively increases the fast component of leakage at the expense of the slow one, in a nonlinear way; in fact an inflection point occurs at around a 1:2 drug/phospholipid ratio. Our explanation of these findings is the presence of two major populations of membrane-incorporated TMX. At a low TMX/phospholipid ratio (below 1:2 drug/phospholipid), where TMX–POPC interactions predominate, TMX distributes mostly homogeneously within the membrane, introducing a perturbation which results in the slow leakage component (k_1_ is the rate constant of this process). Upon further increasing the drug concentration above that ratio, TMX–TMX interactions become predominant, and the drug begins to cluster, which makes the membrane more permeable and allows a faster leakage (k_2_ is the rate constant of this process). This hypothesis was strongly supported by our MD findings showing that the energy of interaction of TMX–POPC began to decrease above a certain TMX/POPC ratio (Figure 7), indicating an increase in TMX–TMX interactions, which could be due to an increased proportion and/or size of clusters as the drug/phospholipid ratio rises. We have shown before that the presence of TMX clusters affects lipid packing and modifies membrane curvature more than isolated drug molecules [11], thus it seems reasonable to assume that permeation of CF across, or near, these two regions of the membrane, should occur at different rates. These TMX-containing regions would constitute membrane defects, allowing permeation of water-soluble molecules, like CF, due to a reduction in membrane compactness and stability, and thus increased permeability, as a consequence of drug insertion. This rationale would also imply that an enhanced drug clustering tendency would result in faster membrane permeability for water soluble compounds. In fact, we have shown before, in fluid DPPC membranes, a location of TMX all along the bilayer and a high clustering tendency, whereas HTMX, due to the presence of the polar −OH group, was mostly found at the acyl chain region, closer to the polar part of the membrane, and displayed a lower clustering tendency [11]. This different location and clustering behavior could well explain why TMX promotes membrane leakage to a larger extent than HTMX.

The simple POPC model membrane used so far, was made more complex by incorporation of other phospholipids commonly found in biological membranes, in order to check the influence of lipid composition on membrane permeabilization. We found that the inclusion of POPG into POPC partially decreased TMX-induced leakage, whereas POPE displayed a more intense effect, essentially abolishing membrane permeabilization. It should be mentioned that PE has been reported before to abolish leakage in various other systems, including peptide-induced [17,18], or even glycolipid and lipopeptide biosurfactant-induced [19,20,21] leakage processes.

FTIR is a valuable tool to investigate lipid–lipid interactions in phospholipid membranes. We have obtained FTIR data showing that incorporation of TMX into the various membranes under study, results in an additional disordering of the hydrophobic region of the membrane bilayer, due to an increase in *gauche*/all*trans* conformer ratio, and a dehydration of the polar interface, which is in agreement with previous data [11,14]. However, the MD data did not show this dehydration effect. This apparent disagreement between the experimental FTIR data and MD simulations could be attributed to the fact that both methods are not measuring the same processes. Thus, whereas the shift in the maximum of the ν_C=O_ band has traditionally been correlated with the hydration of the polar region of the bilayer that not only involves water hydrogen bonding [49], MD simulations provide information exclusively on the ability of phospholipid C=O groups to form hydrogen bonds with water. No substantial differences were observed between the various compositions studied, leading us to discard the idea that potential changes in membrane ordering or hydration might play a relevant role in drug-induced contents leakage. These data were supported by MD simulations, which also showed no differences in *gauche* conformers and C=O water hydrogen bonding (Appendix A) among the various compositions. Therefore, the inhibition of TMX-induced membrane permeabilization by POPG or POPE must be due to other effects, and we found in our MD simulations data a feasible explanation. As we have shown above (Figure 9), the main effects of incorporation of PG or PE, were to considerably enhance the electrostatic interactions at the polar headgroup region, since the entire lipid species used had the same acyl chains and only varied in the headgroup, and to make hydrogen bonding possible. These two effects contribute to increased membrane cohesion in the presence of these two phospholipids, making these membranes less susceptible to druginduced permeabilization.

Finally, we also found that, in mixed PC/PE systems, TMX displays a preferential association with the PC component. This interaction is compatible with the increase in membrane compactness introduced by PE, that would exclude TMX from PErich areas. In fact, a stabilization of the surface state of the phospholipid membrane, due to the smaller headgroup of PE [17], would reduce the deep insertion of the drug and subsequent formation of membrane defects. Furthermore, the interlipid hydrogenbonding initiated by the amine moiety of PE, might also contribute to an enhanced bilayer stability [65], which is supported by our MD simulations showing a reduction in the disordering introduced by the drug at this level of the membrane.

## 5. Conclusions

We have studied the interaction of two drugs currently used for the management of breast cancer, TMX and HTMX, with model phospholipid unilamellar vesicles of various compositions. The results presented in this work have shown that TMX can induce rapid and extensive permeabilization of model phospholipid membranes, in a concentration-dependent manner. The TMXinduced leakage process is biphasic, with a faster and a slower component, whereas in the case of the metabolite HTMX, only the slow component takes place. Biophysical data suggest that TMX induces two types of membrane defects: the slow leakage component is due to the drug, which is homogeneously distributed within the bilayer, whereas the fast one is caused by aggregated TMX molecules or clusters. Finally, it was found that membrane composition can modulate drug-induced leakage, particularly the presence of PE, which is related to the compactness of the bilayer, and results in a preferential association of TMX with the PC component in mixed PC/PE systems. These results add to the previous set of works devoted to studying these two relevant compounds, and should contribute to understanding the potential effects of TMX on biological membranes.

## Figures and Tables

**Figure 1 membranes-13-00292-f001:**
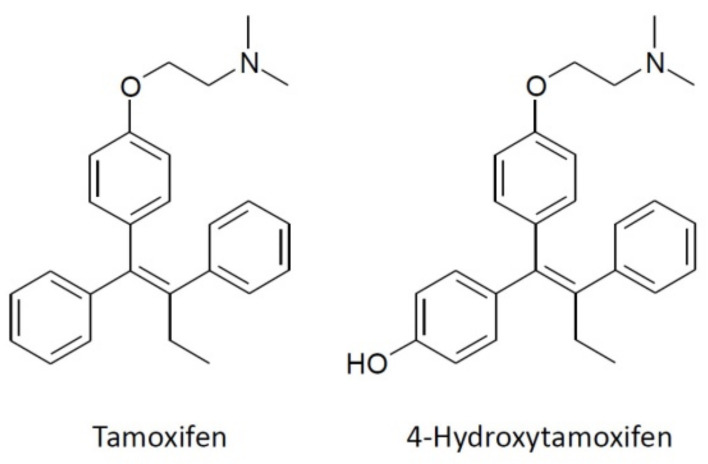
The chemical structures of tamoxifen and 4-hydroxytamoxifen.

**Figure 2 membranes-13-00292-f002:**
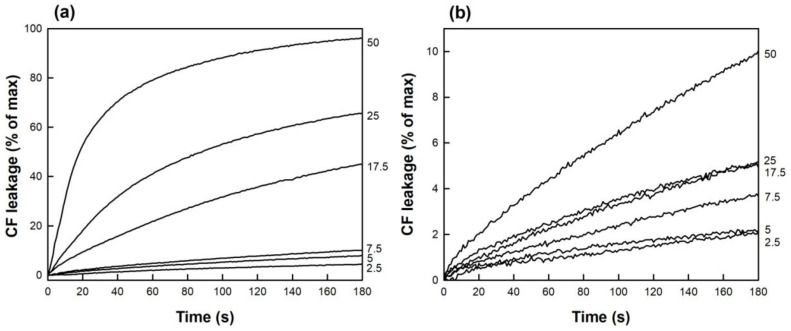
CF leakage curves from POPC LUV upon addition of (**a**) TMX or (**b**) HTMX. Numbers on the right of the traces indicate drug concentrations in μM. Total phospholipid concentration was 20 μM, and temperature was thermostated to 25 °C.

**Figure 3 membranes-13-00292-f003:**
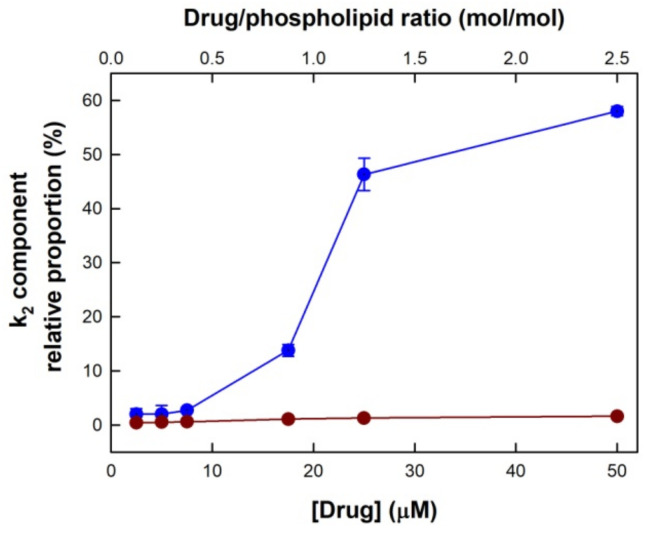
The relative proportion of the k_2_ rate constant component for the druginduced CF leakage from POPC LUV, as a function of TMX (blue) or HTMX (red) concentration. Top axis shows the drug/phospholipid molar ratio. S.E. bars are shown when bigger than the symbols.

**Figure 4 membranes-13-00292-f004:**
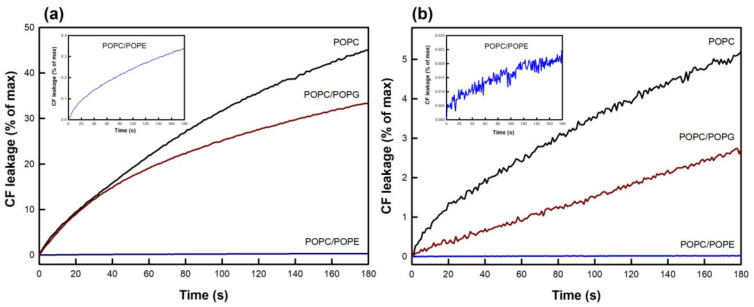
The effect of lipid composition on druginduced CF leakage. (**a**) TMX or (**b**) HTMX were added, at 17.5 μM concentration, to LUV composed of POPC (black), POPC/POPG 5:1.7 (red), or POPC/POPE 5:1.7 (blue). The insets show a close-up view of the curves corresponding to the POPC/POPE composition. Total phospholipid concentration was 20 μM (drug/phospholipid ratio 0.9), and temperature was thermostated to 25 °C.

**Figure 5 membranes-13-00292-f005:**
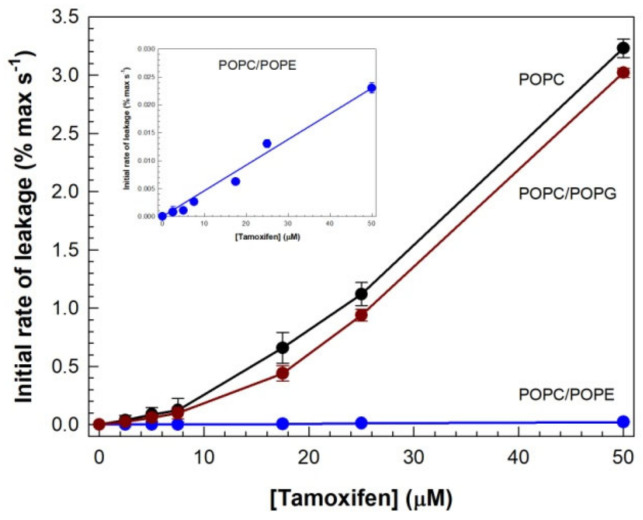
The initial rates of TMXinduced CF leakage as a function of TMX concentration. Initial rates were determined from the curves shown in Figure 3, for LUV composed of POPC (black), POPC/POPG 5:1.7 (red), or POPC/POPE 5:1.7 (blue). The inset shows a close-up view of the plot corresponding to the POPC/POPE composition. S.E. bars are shown when bigger than the symbols.

**Figure 6 membranes-13-00292-f006:**
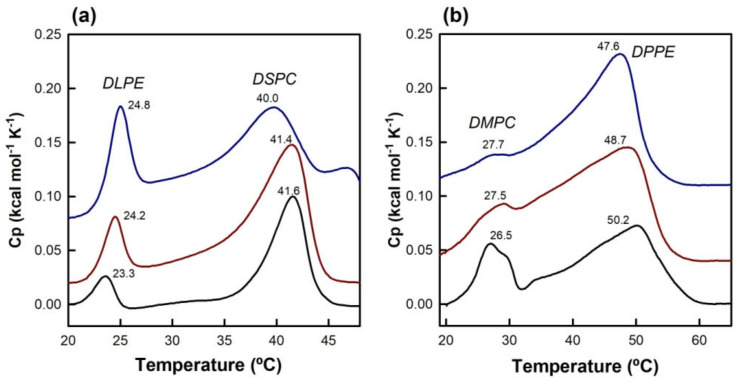
DSC thermograms for systems composed of (**a**) DLPE/DSPC 1:1 and (**b**) DMPC/DPPE 1:1, in the absence and presence of TMX. Black: pure phospholipids; red: +10 mol% TMX; blue: +20 mol% TMX. Total phospholipid concentration was 2 mM, and scanning rate was 4 °C min^−1^. The dashed lines illustrate the procedure used to obtain the transition completion temperatures. Black numbers on the peaks correspond to the T_m_ of the corresponding transition.

**Figure 7 membranes-13-00292-f007:**
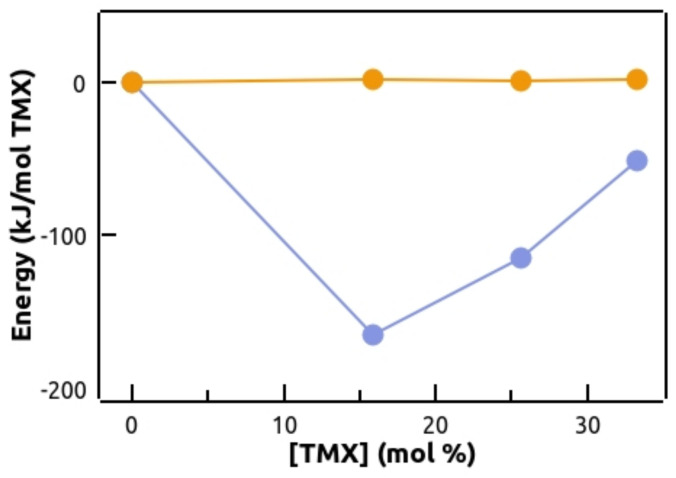
POPC-TMX electrostatic (orange) and hydrophobic (blue) energies of interaction in POPC membranes, as a function of TMX concentration (mol%). Error bars (representing the SD of the trajectories analyzed) are shown when bigger than the symbols.

**Figure 8 membranes-13-00292-f008:**
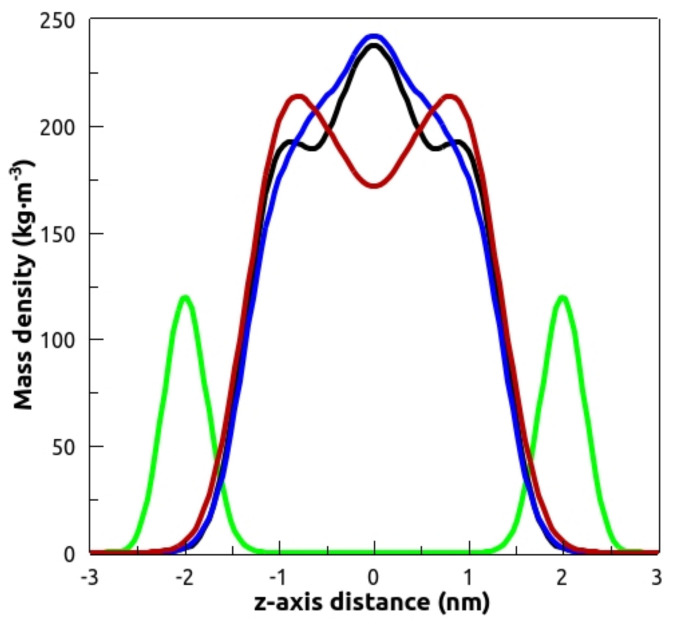
Mass density profiles along the zaxis of the simulation box of the simulated systems: POPC/TMX (phosphorus atoms in green and TMX in black), POPC/POPE/TMX (TMX in blue), and POPC/POPG/TMX (TMX in red). Curves are symmetrized around the center of the bilayer.

**Figure 9 membranes-13-00292-f009:**
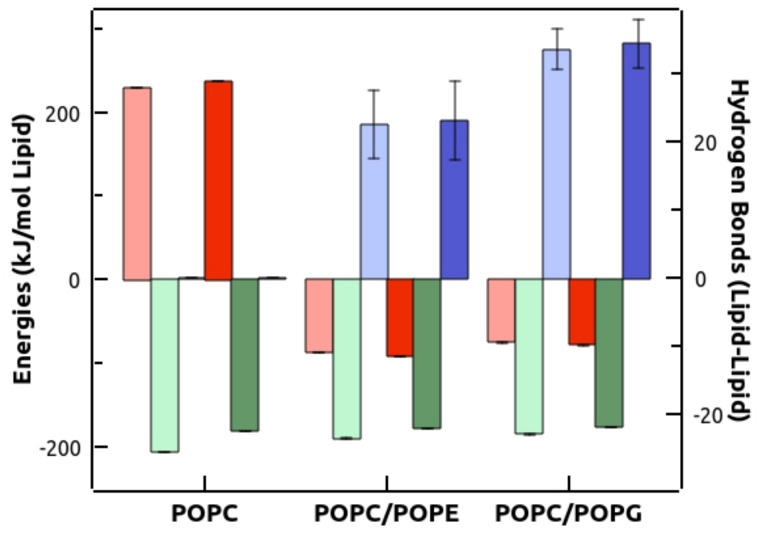
Phospholipid–phospholipid energies of interaction and hydrogen bonding for various lipid compositions, in the absence (light colors) and presence (dark colors) of TMX at 25.6 mol%. Left axis: electrostatic (red) and hydrophobic (green) interactions. Right axis: hydrogen bonds (blue). Error bars represent the SD of the trajectories analyzed.

**Table 1 membranes-13-00292-t001:** Rate constants, and their relative proportions, for TMX- and HTMXinduced CF leakage from POPC unilamellar vesicles, as determined by fitting the data shown in Figure 2 to Equation (1). Data correspond to the mean of three independent experiments ± SD.

[TMX] (μM)	k_1_ (s^−1^) ± SD	% ± SD	k_2_ (s^−1^) ± SD	% ± SD
2.5	1.4 × 10^−4^ ± 0.62 × 10^−4^	98.0 ± 0	0.035 ± 0.0070	2.0 ± 0
5	1.4 × 10^−4^ ± 0.51 × 10^−4^	97.7 ± 0.35	0.035 ± 0.0084	2.3 ± 0.35
7.5	5.5 × 10^−4^ ± 0.71 × 10^−4^	96.8 ± 0.70	0.036 ± 0.0077	3.2 ± 0.70
17.5	25 × 10^−4^ ± 1.1 × 10^−4^	86.8 ± 0.85	0.028 ± 0.0028	13.2 ± 0.84
25	31 × 10^−4^ ± 5.6 × 10^−4^	59.9 ± 8.70	0.023 ± 0.0028	40.1 ± 8.76
50	115 × 10^−4^ ± 20.5 × 10^−4^	45.6 ± 5.10	0.047 ± 0.0160	54.4 ± 5.09
**[HTMX] (μM)**	**k_1_ (s^−1^) ±** SD	**% ±** SD	**k_2_ (s^−1^) ±** SD	**% ±** SD
2.5	0.89 × 10^−4^ ± 0.50 × 10^−4^	99.8 ± 0.22	0.042 ± 0.0082	0.2 ± 0.025
5	0.72 × 10^−4^ ± 0.68 × 10^−4^	98.5 ± 0.75	0.028 ± 0.0035	1.5 ± 0.120
7.5	2.4 × 10^−4^ ± 0.90 × 10^−4^	99.5 ± 0.84	0.049 ± 0.0078	0.5 ± 0.040
17.5	2.2 × 10^−4^ ± 0.44 × 10^−4^	98.6 ± 0.90	0.035 ± 0.0028	1.4 ± 0.008
25	2.7 × 10^−4^ ± 0.72 × 10^−4^	98.7 ± 0.88	0.045 ± 0.0014	1.3 ± 0.006
50	4.8 × 10^−4^ ± 0.62 × 10^−4^	98.9 ± 0.94	0.050 ± 0.0063	1.1 ± 0.004

**Table 2 membranes-13-00292-t002:** Rate constants, and their relative proportions, for TMX- and HTMXinduced CF leakage from unilamellar vesicles of various compositions, as determined by fitting the data shown in Figure 3 to Equation (1). Drug concentration was 17.5 μM. Data correspond to the mean of three independent experiments ± SD.

**Tamoxifen**				
**Composition**	k_1_ (s^−1^) ± SD	% ± SD	k_2_ (s^−1^) ± SD	% ± SD
**POPC**	25 × 10^−4^ ± 1.1 × 10^−4^	86.8 ± 0.85	0.028 ± 0.0028	13.2 ± 0.84
**POPC/POPG**	18 × 10^−4^ ± 1 × 10^−4^	85.7 ± 0.98	0.036 ± 0.0052	14.3 ± 0.98
**POPC/POPE**	9.0 × 10^−4^ ± 1.4 × 10^−4^	100 ± 0		
**4−Hydroxytamoxifen**				
**Composition**	k_1_ (s^−1^) ± SD	% ± SD	k_2_ (s^−1^) ± SD	% ± SD
**POPC**	2.2 × 10^−4^ ± 0.44 × 10^−4^	98.6 ± 0.90	0.035 ± 0.0028	1.4 ± 0.008
**POPC/POPG**	1.9 × 10^−4^ ± 0.32 × 10^−4^	100 ± 0		
**POPC/POPE**	2.0 × 10^−4^ ± 0.50 × 10^−4^	100 ± 0		

**Table 3 membranes-13-00292-t003:** The effect of TMX on the maximum frequency of the ν_CH2_ symmetric stretching and ν_C=O_ stretching bands for various phospholipids and phospholipid mixtures. The molar ratios are given between brackets.

Composition	ν_CH2_ (cm^−1^)	ν_CO_ (cm^−1^)
**POPC**	2847.1	1723.2
**POPC/TMX (5:1.7)**	2852.3	1730.7
**POPC/POPE (5:1.7)**	2847.8	1723.0
**POPC/POPE/TMX (5:1.7:2.3)**	2852.4	1730.7
**POPC/POPG (5:1.7)**	2852.0	1729.2
**POPC/POPG/TMX (5:1.7:2.3)**	2853.0	1732.1

## Data Availability

Not applicable.

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
