# Peer review of "On the Mechanism of Membrane Permeabilization by Tamoxifen and 4-Hydroxytamoxifen"

_membranes, 2023, doi:10.3390/membranes13030292_

Round 1

Reviewer 1 Report

The paper aims to explain the mechanism of action of tamoxifen (TMX), currently used for complementary endocrine treatment of patients with breast cancer, and  its metabolite HTMX, upon model membranes mimicking cell membranes. Altough the theme is quite interesting and experiments were well designed and carefully done, the authors should care with some issues and conclusions before publication, as follows:

1)      The authors present the experimental results (membrane leakage, FTIR, DSC) by examining how TMX and HTMX impact on membrane properties depending on lipid composition. However, the use of PG-containing PC membranes is not well justified. This because: i) there is no PG in the plasma membrane of cancer cells; ii) although the authors justify the use of PG based on its large polar head volume for comparison purposes,  this was not considered in the data discussion and/or conclusions; iii) further,  the presence of PG in the model membranes does not prevent membrane leakage at significant extent as does PE (which is biologically relevant). Therefore, I would suggest to the authors to exclude the PG results, unless its investigation is better justified and discussed in the text.

2)      I do not understand why the authors do not present MD results with HTMX. Indeed, the presence of (OH) groups must favor the drug location towards the hydrophobic region/polar head interface unlikely TMX that mainly resides in the membrane hydrophobic environment as shown by MD results. Maybe, such results could explain, in a deeper way, why TMX promotes membrane leakage to a larger extent than HTMX.

3)      How is the binding or partition of both drugs in the membrane? I suppose TMX binds to a larger extent to the membrane than HTMX. This based on the kinetic constants. Is there any previous data in the literature concerning binding?  I do miss such comparative information in the text.

4)      According to MD results (Fig 9, and lines 378 – 381), the authors did not observe significant differences in water hydrogen bond formation between the C=O group and the water molecules: i) From Fig 9 its seems that the amount of hydrogen bonds are different on PC membranes than PC/PE membranes; ii) if TMX does not impact on water hydrogen bonding, how could one explain the results presented in Table 3 referred to C=O stretching mode (dehydration)?

5)       Uncertainties displayed in the Tables: the way the authors presented the errors in the Tables seems not common. I would suggest to the authors to remake the tables by using the scientific notation for significant digits as, for instance,:

Table 2: k1, SD;  %, SD, respectively to : (26.0+-0.7)x10-4; (86.0+-0.2)

Author Response

1) As suggested by the reviewer, we have included a paragraph in the Discussion section (pp. 14) explaining the biological relevance of PG, and why it was selected for this work.

2) Certainly, TMX and HTMX locate differently within the phospholipid bilayer. This was already shown in our previous article (ref. 11), where we presented a comparative study on the effect of TMX and HTMX on gel and fluid DPPC bilayers. This was briefly commented in the first manuscript and now, as suggested by the reviewer, this discussion has been further extended (pp. 14, bottom).

3) We have included a reference that studied partitioning of TMX and HTMX into DMPC and sarcoplasmic reticulum membranes. Although this is not the same system as ours, it gives an idea on the similar partitioning of both drugs, and the effect of membrane composition and temperature (pp. 13).

4) Figures 9 and S2 show MD results on the effect of lipid composition on different hydrogen bonding. The lines 377-381 refer to hydrogen bonding of C=O groups with water molecules, results which are shown in Fig. S2, whereas Fig. 9 shows phospholipid-phospholipid hydrogen bonding, which is discussed in lines 361-371. However our FTIR data clearly showed that TMX dehydrated the membrane for all the compositions. We did not mentioned MD data on water hydrogen bonding because it showed insignificant differences by effect of TMX, for either composition. We have mentioned this in pp. 13 Results, and further discussed it in pp. 14, providing a possible explanation for this apparent discrepancy.

5) The presentation of the table data has been modified as indicated by the reviewer.

Reviewer 2 Report

In their original research article, Julia Ortiz and colleagues evaluate the permeabilization of membrane models by tamoxifen and 4-hydroxytamoxifen by following carboxyfluorescein fluorescence intensity. The experimental procedures are well designed, and the blanks were properly conducted. Although, the interaction of TMX with lipid membranes has been the subject of many studies, the Authors are successfully at showing a dependence of membrane permeabilization kinetics with compound:lipid ratio, however, as I will detail in my broad comments, the Authors employed compound:lipid proportions that are too high. The Authors were also able to show that the incorporation of POPG or POPE in the lipid membrane slows down the permeabilizing activity of TMX and HTMX and, based on their data, provide a reasonable justification for this observation. The article is well written and organized, however prior to publishing the Authors should address several concerns in order to improve the readability, impact and consistency of their work.

Broad comments

11)     Some important works are not reviewed/cited in the introduction nor used in the discussion:

a.      José B. A. Custódio, Leonor M. Almeida and Vitor M. C. Madeira; A reliable and rapid procedure to estimate drug partitioning in biomembranes; Biochemical and Biophysical Research Communications, Volume 176, Issue 3, 15 May 1991, Pages 1079-1085, where the partition coefficient of TMX and HTMX for lipid membranes containing or not cholesterol is determined.

b.      M Suwalsky , P Hernández, F Villena, F Aguilar, C P Sotomayor; Interaction of the anticancer drug tamoxifen with the human erythrocyte membrane and molecular models; Z Naturforsch C J Biosci. 1998 Mar-Apr;53(3-4):182-90. doi: 10.1515/znc-1998-3-407., where the interaction of TMX with PC and PE containing membranes is characterized in detail. In this work the Authors had already reported that the interaction of TMX with PE membranes in hindered.

22)     The Authors must mention what is the final lipid concentration upon hydration of the lipid film in sections 2.2. and 2.5.. Only in the legend of Figures 2, 4 and 6 it is explicitly mentioned.

33)     How many replicates were performed for membrane permeabilization studies? Statistics is missing. With the information provided the reader does not know if the results, in terms of k1 and k2, for example, are reproducible or not. This is valid for the 3 lipid systems studied: POPC, POPC/POPG and POPC/POPE.

44)     In this kind of compound-membrane interaction studies, the compound concentration is indeed very important. However, compound:lipid ratio is also crucial, and, in this work, the Authors essayed molar ratios as high as 2.5:1, which is a massive compound-to-lipid proportion. Even 1:10 is an already high compound:lipid molar ratio. Molar proportions as high as the ones tested by the Authors are not biologically relevant and the Authors should address this issue in their discussion. This is especially important since only at the highest molar proportions the contribution of k2 becomes more pronounced. I would not be giving too much importance to the data regarding high compound:lipid molar ratios (above 1:2), since, due to the reason mentioned above, they lose relevance.

55)     At 50 µM of TMX, k1 and k2 have the same order of magnitude, 0.0129 and 0.036, respectively. Does it still make sense to make a distinction between a slow and a fast process when they are so close to each other?

66)     This is line with the previous comment: for TMX, k1 ranges from 0.64 x 10-4 to 129 x 10-4 s-1, which represents a range of two orders of magnitude. Is it the process that is taking place with k = 0.64 x 10-4 s-1 the same as the one that takes place with k = 129 x 10-4 s-1?

77)     The Authors state “The values and contributions of k1 and k2 for HTMX-induced CF leakage were essentially not affected by incorporation of either POPG or POPE.”, however it is not the case since for POPC a k2 was determined (though with a very small contribution), whereas for POPC/POPG and POPC/POPE only 1 exponential was used to fit the data.

88)     Lines 326-328: “These results clearly indicated a preferential interaction of TMX with the phosphatidylcholine component of the membrane, irrespective whether it was the most fluid or not, which pointed to the polar headgroups as the principal points of interaction.” – At this point of the manuscript it seems premature to draw such a conclusion and my advice is to not present it. In fact, in their discussion (lines 448-454) the Authors formulate an improved and more explanatory hypothesis regarding these results.

99)     Regarding phospholipid-phospholipid energies of interaction and hydrogen bonding shown in Figure 9, it is my understanding that the data in the absence of TMX should be presented as well.

110)  I ask the Authors to not use the term pores even if they use it as “pores” since it can be misleading even considering that the Authors ask the readers to not consider it literally. The Authors may replace it, for example, by the term “membrane defects”.

111)  Minor English spellcheck is necessary.

Specific comments

The Authors state that cuvettes with a path length of 1 cm were used. Is it 1 cm in both excitation and emission paths? Thus, were 1 x 1 cm cuvettes used? The Authors need to clarify.

Author Response

1) The work by Custodio et al., is now been cited in pp. 13 and pp. 5 (top), and the work by Suwalsky et al. is referred in pp. 13.

2) The final phospholipid concentrations are mentioned in sections 2.2 and 2.5.

3) Three replicates were performed for the leakage experiments, and the results shown in the manuscript correspond to a representative one. We have now calculated the mean of the three experiments and Tables 1 and 2 have been changed accordingly.

4) We have included a paragraph at the beginning of the Results section (pp. 5) addressing the issue of drug concentration. As explained there, if only those drug membrane concentrations expected to be reached in vivo, assuming a homogeneous distribution of the compound, were used in model biophysical studies, the effects observed would be negligible. Nevertheless, the use of higher concentrations does not limit the biological relevance of these studies, since they show effects that can occur locally, where higher concentrations can be reached also in vivo. Some references are included in which this range of drug concentration, or even much higher, is used.

5) Certainly these two data are the same order of magnitude, but k2 is 3 times larger than k1 and, more important, this curve fits much better to two components than to one, the proportion of the two components being very close. The new average values and SD indicate that these differences are significant.

6) The relevant finding that we want to show is that there are two processes taking place at two different rates, with rate constants k1 and k2, and that, upon increasing drug concentration, the faster component becomes more prominent. Certainly the slow process also accelerates at high TMX concentrations and tends to approach the rates of the fast one, due to, as discussed in the text, the progressive increase in drug clustering.

7) The Reviewer is right, and this sentence has been rewritten.

8) As advised by the Reviewer this premature explanation has been removed.

9) As requested, data in the absence of TMX have been now incorporated in Fig. 9.

10) The term 'pore' has been changed by 'membrane defects' throughout the manuscript.

11) Minor English errors have been amended.

Under MM it is now said that it is 1 x 1 cm cuvettes.

Reviewer 3 Report

In this study, Julia Ortiz, José A. Teruel, Francisco J. Aranda, and Antonio Ortiz explored the mechanism of membrane permeabilization by tamoxifen and 4-hydroxytamoxifen through differential scanning calorimetry and molecular dynamics (MD) simulation. By analyzing the preferential association of tamoxifen (TMX) with phosphatidylcholine (PC) in PC/phosphatidylethanolamine (PE) membranes, they discovered that tamoxifen, a medicine used to treat breast cancer, can cause some biological membranes to become permeable, while others are protected.
The title of the manuscript accurately reflects its contents, which demonstrate how tamoxifen affects membrane structure. The writing is good and could be suitable for publication in Membranes with some revisions. The authors should provide more detail in their explanation of the experiments and discussion, and they should compare their work to relevant literature. Furthermore, they should include more information about the effects of drug-like molecules such as fenamates that are not related to the target receptor. It has recently been shown that the action of anti-inflammatory drug compounds may be due to a non-cyclogenase mechanism via the lipid membrane [https://doi.org/10.1016/j.molliq.2022.120502]. It is necessary to add a comparison of your results with the literature to the text of the manuscript.
minor remark
A Methods section should be expanded to include more details about MD calculation, such as an explanation of the TIP3 model and why only 20 ns is used.
Overall, the study provides a detailed analysis of the effects of tamoxifen on membrane structure and permeability. It is well written and could be suitable for publication in Membranes with some revisions.

Author Response

- As suggested, we have commented the indicated reference, and others, on the membrane effects of other drug-like molecules, like the fenamates (pp. 2; pp. 12).

- As requested, three references describing the TIP3P model have been included, and an explanation on why initial 20 ns are used is provided, with reference to Fig. S1.

Round 2

Reviewer 1 Report

The authors did the appropriated revisions. 

Reviewer 2 Report

The Authors have provided highly satisfatory replies to my comments and changed the manuscript accordingly. Considering this the manuscript can be published in the present form.